Microbiology
Spectrum

# Evaluation and comparison of three qPCR commercial assays and three automated platforms for the detection of monkeypox virus DNA

Marie Usal,[1] Julien Andréani,[1,2] Aurélie Truffot,[1,3] Capucine Chevalier,[1] Sylvie Larrat,[1] Julien Lupo,[1,3] Patrice Morand,[1,3] Olivier Ferraris,[4] Raphaële Germi[1,3]

ABSTRACT   This study compared the analytical performance of three commercial real-time PCR assays for the detection of monkeypox virus (MPV) DNA, on their own platforms and/or on open platforms, were evaluated at the virology laboratory of Grenoble-Alpes University Hospital (CHUGA): the VIASURE monkeypox virus real-time PCR Reagents on the BD MAX System; the MPXV AMP Kit on the Alinity m system; and the Altona RealStar Zoonotic *Orthopoxvirus* PCR Kit 1.0 on the Argene System (EMAG, ESTREAM [Biomérieux], and LC480 [Roche]). Results obtained on these platforms were compared to those from the French National Reference Center (NRC) for orthopoxviruses. A total of 52 samples were selected by the NRC and the CHUGA: 8 dilutions of MPV culture supernatant, 2 supernatants from cowpox and vaccinia virus cultures, 32 clinical samples (20 positive/12 negative), and 10 external quality control samples from the "Quality Control for Molecular Diagnostics" (QCMD) program. The sensitivity and specificity of the three assays, compared to the NRC assay, were 100% on the samples tested. All positive samples were correctly identified as positive with all three assays, and all negative samples were negative for MPV detection. Overall, the cycle threshold (Ct) values obtained with the VIASURE, Abbott, and Altona assays were lower than those obtained with the NRC PCR in 84.38%, 81.25%, and 75.86% of positive samples, respectively. In conclusion, all three assays demonstrated good analytical performance and were suitable for MPV detection.

IMPORTANCE   Mpox is an emerging zoonotic disease caused by the monkeypox virus, mostly reported in Africa. In 2022, an outbreak characterized mainly by anogenital rash in MSM (men having sex with men) occurred in Europe. In 2024, many cases of infection were reported, presenting with generalized skin rashes. Detection of monkeypox virus DNA in biological samples is crucial for diagnosis, interruption of transmission chains, treatment initiation, and vaccination. During the European outbreak in 2022, numerous companies rapidly developed qPCR kits. In France, the performance of these kits has been evaluated by the French National Reference Center (NRC) for orthopoxviruses. The Grenoble-Alpes University Hospital collaborated with the NRC to evaluate three available assays on its own platforms and/or on open platforms available in its virology laboratory. All three assays demonstrated good analytical performance and can be used with confidence.

KEYWORDS   diagnosis, monkeypox virus, MPV, real-time PCR, analytical performance

**Peer Reviewer** Gundallahalli Bayyappa Manjunathareddy, ICAR-National Institute of Veterinary Epidemiology and Disease Informatics, Bengaluru, Karnataka, India

Address correspondence to Raphaële Germi, rgermi@chu-grenoble.fr.

Marie Usal and Julien Andréani contributed equally to this article. Marie Usal supervised the tests, analyzed the results, and wrote the article; Julien Andreani performed the statistical analysis, created the graphs, and wrote the paper with Marie.

The authors declare no conflict of interest.

Monkeypox virus (MPV) is a double-stranded DNA virus belonging to the *Poxviridae* family (genus *Orthopoxvirus*). This virus is endemic to Africa, where it primarily infects rodents. Similarly, cowpox virus is found in rodents in Europe (1, 2). Other

orthopoxviruses are endemic in specific regions, such as vaccinia virus in Brazil (3, 4) and buffalopox virus in India (5, 6). Globally, since 2022, the MPV has caused at least two distinct epidemics: the first, which emerged in 2022, was linked to clade II (clade IIb) (7), and the second, which began in 2023, originated from clade I (clade Ib) (8–10). During the first outbreak, the virus spread worldwide, and it is currently circulating at low levels with only sporadic cases being reported. The second outbreak, associated with the Democratic Republic of Congo (DRC), is currently affecting neighboring countries (Burundi, Central African Republic, Rwanda, and Uganda), and sporadic cases have also been observed in Kenya, Thailand, and Europe. This outbreak has spread worldwide, with currently two major hotspots in the DRC and Madagascar. Sequencing data, obtained after capture enrichment, suggest co-circulation of clades Ia and Ib in the DRC since July 2024 (11). In response to the risk of spread around the world, companies have developed various real-time PCR test kits as rapid diagnostic solutions (12–14). Detection of MPV DNA in biological samples is crucial for diagnosing infection, interrupting transmission chains, initiating treatment, and introducing vaccination (15, 16). Furthermore, data are now available on MPV carriage and detection in various biological fluids (17). Indeed, studies have shown that MPV DNA remains detectable for approximately 20 days in saliva and the oropharyngeal area and around 14 days in feces and semen (17–19). In blood, DNAemia appears to be very low and is usually detectable only during the first 10 days after symptom onset, thereby reducing the risk of viral transmission through transfusion (20, 21). This knowledge of viral replication is highly valuable for clinical diagnosis. The French National Reference Center (NRC) for orthopoxviruses has recommended the use of a kit targeting the tumor necrosis factor receptor gene (also known as G2R or J2R) (22). Several assays have been evaluated by the NRC (available at https://www.sfm-microbiologie.org/actualites/monkeypox/monkeypox-fiches/monkeypox-rapports/). However, it is not feasible to test all available kits across every automated platform in order to comprehensively assess their performance. This study, conducted by the Grenoble-Alpes University Hospital (CHUGA) virology laboratory in close collaboration with the NRC, aimed to describe and compare the analytical performance of three commercial systems (kits + platforms): the VIASURE monkeypox virus Real-Time PCR Reagents on the BD MAX System; the MPXV AMP Kit on the Alinity m instrument; and the Altona RealStar Zoonotic *Orthopoxvirus* PCR Kit 1.0 on the Argene solution platform (EMAG, ESTREAM [Biomérieux], and LC480 [Roche]). This last kit is designed to detect non-variola orthopoxviruses and, therefore, to identify monkeypox, cowpox, and vaccinia viruses. The results obtained with these commercial systems were also compared to those of the NRC.

## MATERIALS AND METHODS

### Assay design

This study was conducted on a panel of 52 samples, including 32 clinical specimens, 10 culture supernatants, and 10 external quality controls. Among these, 32 tested positive and 20 tested negative for MPV DNA. The number of results obtained varied slightly between commercial systems due to insufficient sample volume in some cases. The analytical performance of the three commercial systems was assessed in terms of sensitivity, specificity, and accuracy by comparing both qualitative results (positive/negative) and semi-quantitative results (cycle threshold [Ct] values) with those obtained by the NRC (Fig. 1).

### Patient, culture, and control samples

The French NRC for orthopoxviruses prepared eight inactivated cell culture supernatants containing different concentrations of MPV DNA (Strain MPXV/France/IRBA2211/2022 ON755039 Clade IIb.B1) (23) for sensitivity testing. It also prepared two additional supernatants containing vaccinia virus (Strain IHD: ATCC VR-156) and cowpox virus

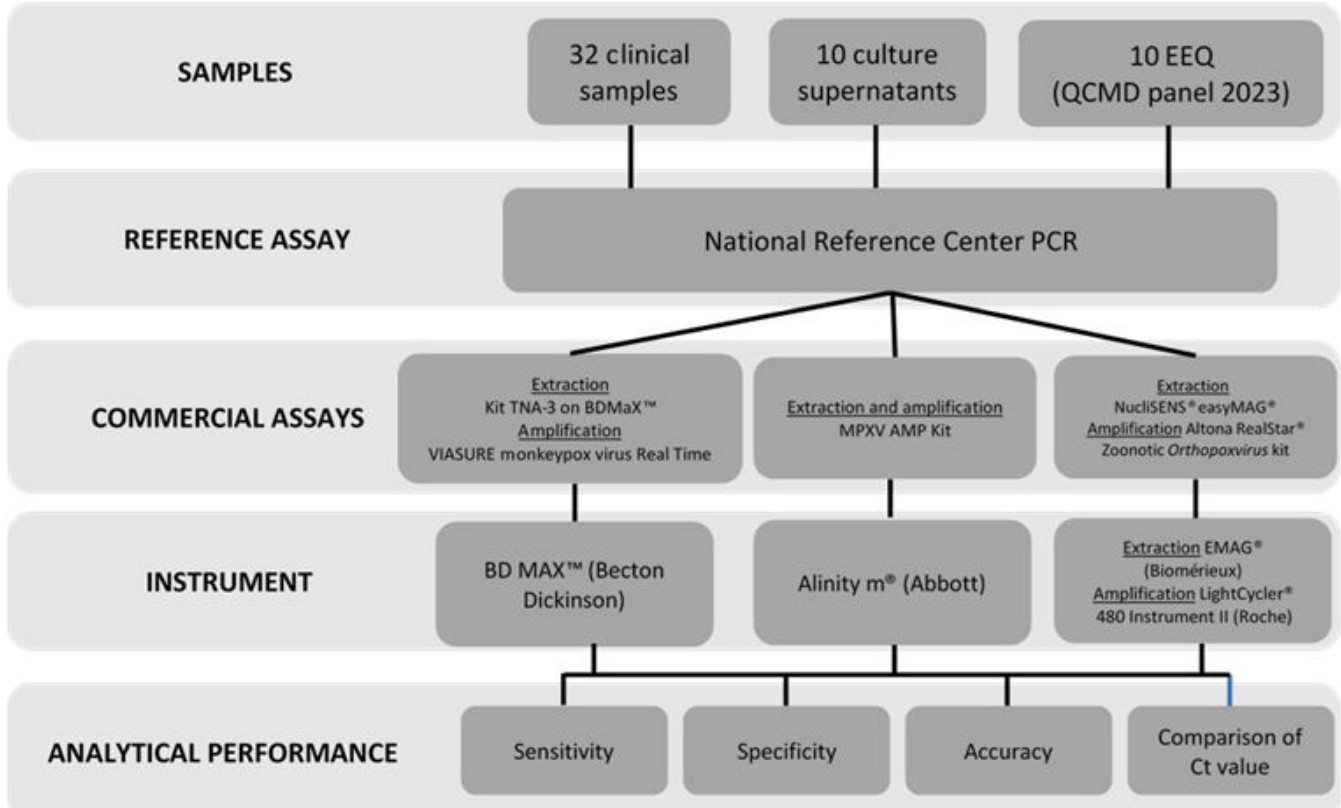

**FIG 1** Evaluation of commercial systems for monkeypox virus DNA detection: study design. EEQ, external quality controls; QCMD, Quality Control for Molecular Diagnostics.

(Strain Brighton: ATCC VR-302) for specificity testing. These supernatants were sent to CHUGA together with a panel of 17 mucocutaneous swabs (6 negative and 11 positive for MPV).

In addition, the CHUGA laboratory selected 15 clinical samples (6 negative and 9 positive for MPV) that had already been analyzed by the NRC in a clinical setting. Among these, four samples were negative for MPV but positive for other viruses (two positive for Varicella Zoster Virus [VZV], one positive for enterovirus, and one positive for Herpes simplex virus type 1 [HSV-1]) to ensure the absence of cross-detection. All clinical samples from CHUGA were skin swabs, except the enterovirus-positive sample, which was a nasal swab collected in viral transport medium (Citoswab).

The 10 external quality controls from the Quality Control for Molecular Diagnostics (QCMD) program, purchased as part of the laboratory's quality assurance system, were used to assess accuracy. These included seven transport mediums positive for MPV DNA (four clade I and three clade IIb) and three transport mediums negative for MPV (one positive for vaccinia virus, one positive for cowpox virus, and one virus-free).

All samples were stored at −80°C until use in this study. Clinical samples were collected, analyzed, and frozen at least 6 months before being used in this study. These samples, therefore, underwent additional thawing compared to routine analysis, as they had external quality controls and culture samples.

## Nucleic acid extraction and real-time PCR

At the NRC, total DNA was extracted from 200 µL of each sample using the Qiagen QIAamp DNA mini kit (manual extraction), and the PCR was performed using the MPV real-time assay described by Ly et al. (22). For this pan-MPV PCR, each reaction consisted of 5 µL extracted DNA and 15 µL iTaq Universal Probes Supermix (BioRad), the Supermix

**TABLE 1** Description of commercial kits and French National Reference Center for orthopoxviruses (NRC) PCR

| Commercial/laboratory | Assay name | Target genes | Controls | Reference |
|---|---|---|---|---|
| BD | VIASURE monkeypox virus Real-Time PCR Reagents RUO and BD MAX TNA-3 (for extraction) | G2R[b] and F3L genes | Cellular control | 444211/442828 |
| Abbott | MPXV AMP Kit | B7R and J2R genes | Cellular and internal controls | 09R06-092 |
| Altona | Altona RealStar *Orthopoxvirus* PCR Kit RUO, including MPV[a] | A3L gene | Internal control | 361003 |
| NRC | PCR and Qiagen QIAamp DNA mini kit (for extraction) | G2R gene[b] | Internal control | (22) |

[a]MPV, monkeypox virus.
[b] G2R gene, tumor necrosis factor gene.

containing 0.4 µM of each primer (Eurogentec: F: GGAAAATGTAAAGACAACGAATACAG; R: GCTATCACATAATCTGGAAGCGTA), 0.2 µM of probe (Eurogentec: Quasar 705—AAGCCGTA ATCTATGTTGTCTATCGTGTCC—BHQ-3), and 1X SPC Mix (Yakima Yellow-BHQ-1, Eurogentec). All assays were performed on a CFX96 thermocycler (BioRad). The data and results were analyzed and reported using BioRad CFX Maestro v2.3 (Table 1).

Three commercial systems were evaluated: (i) the VIASURE monkeypox virus Real-Time PCR Reagents on the BD MAX System (Becton Dickinson, BD), (ii) the MPXV AMP Kit on the Alinity m System (Abbott), and (iii) the RealStar Zoonotic *Orthopoxvirus* PCR Kit 1.0 on the Argene solution consisting of EMAG (Biomérieux) for extraction, ESTREAM (Biomérieux) for distribution of extracts and PCR Mix, and the LightCycler 480 Instrument II (Roche) for amplification and detection (Altona) (Table 1). The RealStar *Orthopoxvirus* kit had been used in the clinical setting at the CHUGA laboratory at the beginning of the MPV outbreak. It detects non-variola *Orthopoxvirus* species, including cowpox virus, MPV, raccoonpox virus, camelpox virus, and vaccinia virus. For this reason, all MPV-positive samples from CHUGA were systematically sent to the NRC for confirmatory analysis of specific MPV infection. By contrast, the BD and Abbott assays are specific for MPV.

The VIASURE monkeypox virus Real-Time PCR Reagents and the MPXV AMP Kit were provided free of charge by BD and Abbott, respectively, for the purposes of this study.

## Analytical performance evaluation

The NRC assay was used as the reference method.

Analytical sensitivity for each assay was defined as the proportion of positive results obtained with the evaluated platforms among samples confirmed positive by the NRC reference assay ($n = 32$ for the BD and Abbott assays and 29 samples for the Altona assay).

Analytical specificity was assessed using culture supernatants and external quality controls (EQC) positive for other *Orthopoxvirus* species (cowpox virus and vaccinia virus), as well as clinical samples positive for viruses causing similar rashes but belonging to other families, such as HSV-1, VZV, and enterovirus.

Accuracy was evaluated using EQC samples and based on reports provided by the QCMD program.

Linearity and the lowest concentration detected for each commercial system were determined using serial dilution ($10^{-2}$ to $10^{-9}$) of MPV culture supernatants. Viral DNA loads in each dilution were quantified by NRC qPCR as follows: $10^{-2} = 7.37E^{+03}$ copies/µL, $10^{-3} = 6.65E^{+02}$ copies/µL, $10^{-4} = 6.05E^{+01}$ copies/µL, $10^{-5} = 8.83E^{+00}$ copies/µL, $10^{-6} = 5.20E^{-01}$ copies/µL.

## RESULTS

### Analytical sensitivity and specificity

All three commercial systems demonstrated on the samples tested an analytical sensitivity of 100%, as every sample identified as positive for MPV DNA by the NRC was also positive with the commercial systems. No false negative was observed.

Similarly, all three systems showed an analytical specificity of 100%, as no MPV DNA was detected in samples classified as negative for MPV but positive for other viruses by the NRC or the CHUGA laboratory. No false positive was observed with any of the three commercial assays (Table 2).

### Accuracy

The three methods successfully detected two distinct MPV clades, West Africa and Congo Basin Region (I and IIb), with low Ct values (Table 3) indicating 100% accuracy for the Abbott and BD kits. The Altona assay also detected two positive samples corresponding to cowpox and vaccinia viruses, as expected because the kit was designed to detect *Orthopoxvirus* species.

### Linearity

The NRC provided CHUGA a series of dilutions of inactivated culture supernatant. The concentration of the MPV genome in each dilution was quantified using the NRC qPCR assay. Each dilution was evaluated with all three assays (Fig. 2). All three commercial systems produced linear results.

This dilution range also allowed the determination of the lowest dilution detected by the three commercial systems. As observed with the NRC method, all three systems were able to detect the MPV DNA from dilutions $10^{-2}$ to $10^{-6}$ ($10^{-2} = 7.37E^{+03}$ copies/µL, $10^{-3} = 6.65E^{+02}$ copies/µL, $10^{-4} = 6.05E^{+01}$ copies/µL, $10^{-5} = 8.83E^{+00}$ copies/µL, $10^{-6} = 5.20E^{-01}$ copies/µL). The lowest concentration (dilution $10^{-7}$) was not detected by any PCR system.

TABLE 2   Analytical specificity of three commercial assays for the detection of monkeypox virus

| Viruses | Viral load or Ct value[f] | Matrix | BD[a] | Abbott[b] | Altona[c] (*Orthopoxvirus*) |
|---|---|---|---|---|---|
| VZV[d] | >8 log copies/mL | Swab | Negative | Negative | N.D.[h] |
| VZV | 3 log copies/mL | Swab | Invalid[l] | N.D. | N.D. |
| HSV-1[e] | 5.9 log copies/mL | Swab | Negative | Negative | N.D. |
| Enterovirus | 25.12 (Ct) | Swab | Negative | Negative | N.D. |
| Negative for VZV and HSV | N.A.[g] | Swab | Negative | Negative | N.D. |
| Negative for VZV and HSV | N.A. | Swab | Negative | Negative | N.D. |
| Vaccinia virus | >8 log copies/mL | Supernatant | Invalid | Error[k] | 18.64[m] (Ct) |
| Cowpox virus | >8 log copies/mL | Supernatant | Negative | Negative | 15.08 (Ct) |
| Vaccinia virus[j] | N.C.[i] | N.C. | Negative | Negative | 28.84 (Ct) |
| Cowpox virus[j] | N.C. | N.C. | Negative | Negative | 23.26 (Ct) |

[a]BD, VIASURE monkeypox virus Real-Time PCR Reagents on the BD MAX System.
[b]Abbott, MPXV AMP Kit on the Alinity m system.
[c]Altona, RealStar Zoonotic *Orthopoxvirus* PCR Kit 1.0 on the Argene System.
[d]VZV, varicella zoster virus.
[e]HSV, herpes simplex virus.
[f]Ct, cycle threshold.
[g]N.A., not applicable.
[h]N.D., not done for insufficient volume.
[i]N.C., not communicated.
[j]Sample from QCMD panel 2023 (Quality Control for Molecular Diagnostics).
[k]Missing data due to error in instrument, no result.
[l]Cellular control not detected.
[m]RealStar Zoonotic *Orthopoxvirus* PCR Kit 1.0.

**TABLE 3** Accuracy of three commercial assays for monkeypox virus detection using the 2023 QCMD panel

| Sample | NRC[a] (Ct[g] value) | BD[b] (Ct value) | Abbott[c] (Ct value) | Altona[d] (Ct value) | QCMD[e] | | |
|---|---|---|---|---|---|---|---|
| | | | | | Result (core EQA[e])[f] | Detection frequency (educational EQA)[f] | Clade |
| Q1 | Negative | Negative | Negative | Negative | Negative | | |
| Q2 | Negative | Negative | Negative | 28.84 | | Negative | Vaccinia virus (strain Elstree) |
| Q3 | 30.1 | 30.8 | 31.21 | 31.8 | Frequently detected | | Clade IIb (West African clade/2022 outbreak isolate) |
| Q4 | 30.56 | 25.7 | 27.67 | 31.12 | Detected | | Clade I (Congo Basin clade/Central African isolate) |
| Q5 | 31.02 | 31.4 | 29.48 | 29.17 | Frequently detected | | Clade IIb (West African clade/2022 outbreak isolate) |
| Q6 | 31.17 | 29.2 | 26.8 | 28.78 | Frequently detected | | Clade I (Congo Basin clade/Central African isolate) |
| Q7 | 33.76 | 32.6 | 31.53 | 33.11 | | Detected | Clade I (Congo Basin clade/Central African isolate) |
| Q8 | 34.73 | 33.1 | 31.95 | 32.05 | | Detected | Clade I (Congo Basin clade/Central African isolate) |
| Q9 | 36.26 | 34.2 | 35.04 | 35 | | Detected | Clade IIb (West African clade/2022 outbreak isolate) |
| Q10 | N.D.[h] | Negative | Negative | 23.26 | | Negative | Cowpox virus (OPV 88-Lunge) |

[a]NRC, French National Reference Center for orthopoxviruses.
[b]BD, VIASURE monkeypox virus Real-Time PCR Reagents on the BD MAX System.
[c]Abbott, MPXV AMP Kit on the Alinity m System.
[d]Altona, RealStar Zoonotic *Orthopoxvirus* PCR Kit 1.0 on the Argene System.
[e]EQA, external quality assessment; QCMD, Quality Control for Molecular Diagnostics.
[f]Consensus result of the peer group determined from the qualitative results reported by participants.
[g]Ct, cycle threshold.
[h]N.D., not done for insufficient volume.

## Semi-quantitative analysis (Ct value)

Ct values were compared for 29 positive samples using the Altona assay and for 32 positive samples using the Abbott and BD systems. Three samples were not tested with the Altona method due to insufficient sample volume.

Ct values obtained with the BD, Abbott, and Altona assays were lower than those obtained with the NRC PCR in 84.38%, 81.25%, and 75.86% of positive samples, respectively. For the positive clinical samples analyzed (P1 to P20), 94.12% of Ct values generated with the Altona assay were lower than those obtained with the NRC PCR, as were 85% of Ct values obtained with the BD assay and 75% of those obtained with the

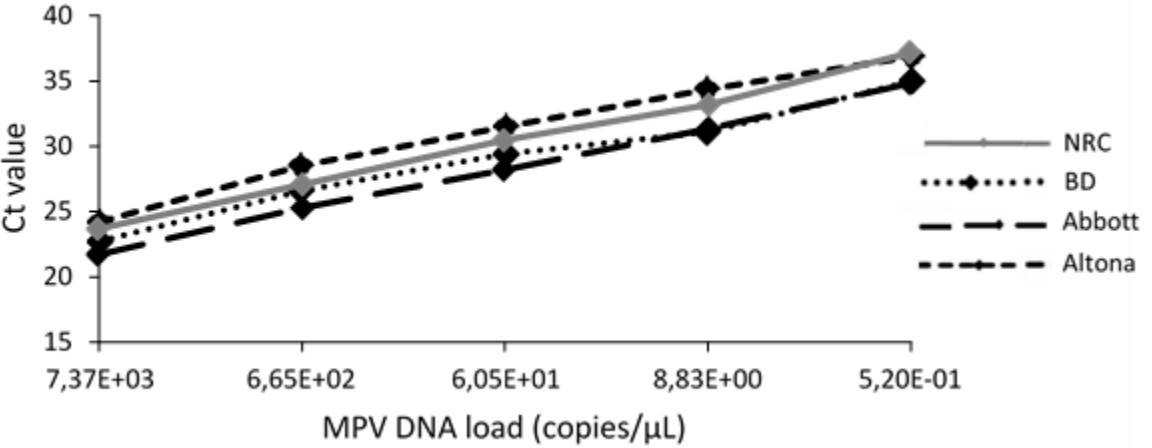

**FIG 2** Cycle threshold (Ct) values obtained from serial dilutions of culture supernatant containing monkeypox virus (MPV) DNA using three commercial systems and the French National Reference Center for orthopoxviruses (NRC) assay. BD, VIASURE monkeypox virus Real-Time PCR Reagents on the BD MAXTM System; Abbott, MPXV AMP Kit on the Alinity m system; Altona, RealStar Zoonotic Orthopoxvirus PCR Kit 1.0 on the Argene solution.

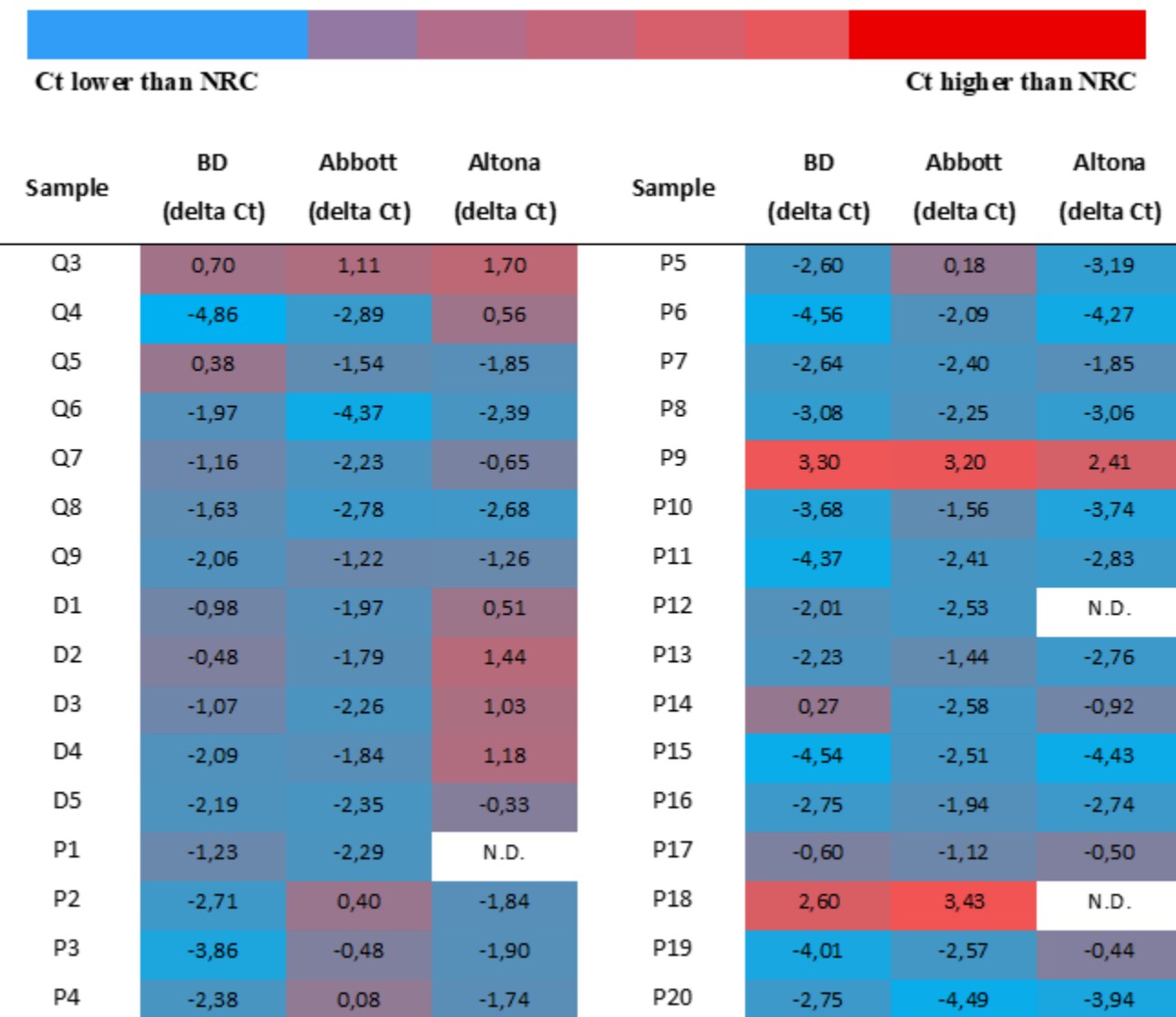

**FIG 3** Comparison of cycle threshold (Ct) values obtained with three commercial systems and the French National Reference Center for orthopoxviruses (NRC) assay on samples positive for monkeypox virus DNA. BD, VIASURE monkeypox virus Real Time PCR Reagents on the BD MAX System; Abbott, MPXV AMP Kit on the Alinity m system; Altona, Real Star Zoonotic Orthopoxvirus PCR Kit 1.0 on the Argene System; Delta Ct, Ct obtained with commercial system − Ct obtained with NRC assay; Q, external quality assessment sample (QCMD, Quality Control for Molecular Diagnostics); D, dilutions of inactivated monkeypox virus culture supernatants; P, patient sample; N.D., not done due to insufficient volume.

Abbott assay (Fig. 3). For the culture supernatant dilutions (D1 to D5), the BD and Abbott assays yielded lower Ct values than NRC method for 100% of the dilution samples, whereas the Altona assay yielded lower Ct values for only 20% (D1 to D4) of the dilutions (Fig. 3). Finally, analysis of external quality control samples (Q3 to Q9) showed that the Abbott assay produced lower Ct values than the NRC method in 85.71% of cases, while the Altona and BD assays did so in 71.43% of cases (Fig. 3).

## Assessment of the robustness of cellular and internal controls

The Altona kit includes an internal control designed to detect PCR inhibitors. The BD assay incorporates a cell control that ensures sample quality and detects PCR inhibitors. The Abbott assay includes both types of controls.

**TABLE 4** Results obtained for samples whose cell control or internal control was not amplified in at least one assay

| Sample | BD[c] MPV[a] (qualitative result) | BD cellular control (Ct)[b] | Abbott[d] MPV (qualitative result) | Abbott internal control (Ct) | Abbott cellular control (Ct) | Altona[e] MPV (qualitative result) | Altona internal control (Ct) |
|---|---|---|---|---|---|---|---|
| P9 | Positive | no Ct value | Positive | 25.86 | 29.24 | Positive | 24.95 |
| P14 | Positive | no Ct value | Positive | 25.32 | 26.8 | Positive | 24.24 |
| P16 | Positive | no Ct value | Positive | 25.99 | 27.23 | Positive | 24.86 |
| P17 | Positive | no Ct value | Positive | 25.6 | No Ct value | Positive | 25.84 |
| P20 | Positive | no Ct value | Positive | 25.24 | 31.51 | Positive | 25.3 |
| P21 | Invalid[f] | no Ct value | Negative | 25.28 | 30.22 | N.D. | N.D. |
| P22 | Invalid | no Ct value | Negative | 28.94 | 28.02 | N.D. | N.D. |
| P24 | Invalid | no Ct value | Negative | 25.29 | 33.08 | N.D. | N.D. |
| P25 | Invalid | no Ct value | Negative | 24.28 | 33.56 | N.D. | N.D. |
| P29 | Invalid | no Ct value | N.D.[g] | N.D. | N.D. | N.D. | N.D. |

[a]MPV, monkeypox virus.
[b]Ct, cycle threshold.
[c]BD, VIASURE monkeypox virus Real-Time PCR Reagents on the BD MAX System.
[d]Abbott, MPXV AMP Kit on the Alinity m system.
[e]Altona, RealStar Zoonotic *Orthopoxvirus* PCR Kit 1.0 on the Argene solution.
[f]Invalid, negative MPV result and negative cellular control.
[g]N.D., not done due to insufficient volume.

Table 4 lists the results obtained for samples whose cell control or internal control was not amplified in at least one assay. The internal control in the Altona and Abbott assays was successfully amplified for all patient samples. The cell control was not properly amplified in 10 samples for the BD kit and one sample for the Abbott kit. Culture supernatants were not included in this table because they do not contain cells and therefore yielded negative cellular control amplification results, as expected. Similarly, the QCMD samples were excluded due to their unknown composition.

Among the 31 patient samples tested with the Abbott system, the cell control of one sample (3.23%) was not amplified. This sample was positive for MPV DNA, so all samples tested yielded valid results.

Among the 32 patient samples tested with the BD assay, the cell control of 10 samples (31.25%) was not amplified. This had no impact on the five samples presenting with MPV DNA-positive results, as the BD assay reported a positive result. On the contrary, for the five samples presenting with MPV DNA-negative results, the BD system reported uninterpretable results (15.63%). It should be noted that the BD assay recommends an input volume of 200–400 µL, and 200 µL was used in this study.

## DISCUSSION

The mucocutaneous symptoms of MPV are similar to those caused by other viruses, such as HSV, VZV, or enteroviruses. Therefore, a definitive etiological diagnosis cannot be based solely on clinical manifestations (24) and requires nucleic acid amplification tests, such as qPCR. qPCR assays are rapid, specific, and sensitive methods that support preventive measures (isolation, vaccination) to limit disease transmission. During the European MPV outbreak in 2022, numerous companies rapidly developed qPCR kits designed for use on their own platforms and/or on open platforms. The performance of most of these kits has been evaluated, and in France, the NRC for orthopoxviruses was responsible for conducting these assessments. The results are available at https://www.sfm-microbiologie.org/actualites/monkeypox/monkeypox-fiches/monkeypox-rapports/. As the NRC cannot possess all the expensive platforms marketed by different companies, the CHUGA virology laboratory proposed a collaboration with the NRC to evaluate three assays on three automated platforms available in the laboratory.

In this study, three commercial assays and their corresponding automated platforms available at CHUGA were evaluated. Since all samples identified as positive for MPV by the NRC method were also positive with the three commercial systems, their sensitivity

was assessed as 100% on the samples tested. In addition, all samples determined to be negative for MPV by the NRC or by CHUGA were also negative with the commercial assays, except for the Altona *Orthopoxvirus* assay, which detected vaccinia virus and cowpox virus as expected. Among the MPV-negative samples, eight were positive for HSV-1, VZV, enterovirus, vaccinia virus, and cowpox virus in order to evaluate potential cross-reactivity. Because the lesions caused by these viruses are similar to those caused by MPV and may occur at the same anatomical sites, differential diagnosis is necessary. None of the samples tested positive for MPV; the specificity of the three systems was assessed to be good. Although the number of samples is not very large, it is consistent with findings reported in other studies. These results were further supported by testing the QCMD panel, which confirmed that all three kits were able to detect both clade I and clade II MPV. Although the instructions for use provided by the manufacturers of the three assays state—based on *in silico* analyses—that both clades can be detected, the QCMD results provide reassuring evidence regarding their reliability during future outbreaks. Analysis of the QCMD panel also confirmed the overall accuracy of the systems. To complete the performance evaluation, a dilution series of MPV culture supernatant was analyzed with each kit to assess linearity. All three assays demonstrated excellent linearity between 7,000 and 0.5 copies of DNA per microliter. This also confirmed that the assays were able to detect extremely low viral loads, comparable to the NRC test.

Most laboratories provide qualitative results, which are generally sufficient for clinicians in routine patient care. Nevertheless, quantitative (copies/mL) or semi-quantitative (Ct value) results may be useful in certain complex situations. We therefore also compared the Ct values obtained with all the systems tested to those obtained with the NRC method. Overall, the Ct values generated by the three commercial methods were lower than those obtained with the reference method, except for the supernatant dilutions for which the Altona assay produced higher Ct values than the NRC assay. This finding highlights that, depending on the sample matrix, assay performance—particularly nucleic acid extraction efficiency—may vary, and that the type of sample should be considered when evaluating or comparing techniques. It is also noteworthy that Ct differences may depend on calibration standards, which can differ between manufacturers. Thus, Ct values are assay-dependent and influenced by multiple factors, and lower Ct values should therefore not necessarily be interpreted as superior analytical performance.

This study had the advantage of evaluating three assays within a single workflow, including two fully integrated systems with dedicated extraction and PCR platforms. A Canadian study published in 2024 compared five assays, including the VIASURE monkeypox virus Real-Time PCR Detection Kit and the RealStar Zoonotic *Orthopoxvirus* PCR Kit 1.0 (25). The main difference from our study was that a single extraction platform (NucliSENS EMAG) was used for all assays, and the two thermocyclers used for DNA amplification were not specified. This suggests that the dedicated platforms were not used. This study assessed the positive percent agreement and the negative percent agreement rather than specificity and sensitivity, which makes comparison difficult. Nevertheless, the authors reported that six samples with Ct values > 32 were not detected. In the present study, all eight samples with Ct value > 32 (three clinical samples, three QCMD samples, and two MPV supernatant dilutions) were detected by all assays. The Altona assay was also evaluated in a 2024 Italian study, which reported results comparable to ours (100% specificity and sensitivity at lower Ct values) (26). A third study published in 2024 by Abbott reported the analytical performance of their automated high-throughput molecular tests, including the limit of detection and linearity. The clinical performance of the Abbott test compared with results from the Centers for Disease Control and Prevention also showed 100% agreement (22).

It is noteworthy that other commercial assays have also been evaluated in the literature (12–14, 26, 27). They are therefore not included in the above discussion, but several points can be drawn from these studies, i.e., (i) swabs from vesicular lesions

appear to be more sensitive than oropharyngeal samples (27); (ii) qPCR assays can target various genes (such as G2R, F3L, …) and demonstrate good sensitivity and specificity (22, 25, 26).

The use of a dedicated platform offers the advantage that the entire analytical workflow (virus and controls) is performed on board, meaning that even minor defects in the PCR process immediately result in an uninterpretable outcome. In our study, five uninterpretable results were obtained with the BD assay. A detailed examination of the control data revealed that these were attributable to a failure of the cellular control to be amplified in MPV-negative samples. Notably, none of the previously cited studies reported internal control performance, with the exception of the Canadian study, which mentioned an uninterpretable result with the BD assay but provided no explanation. The BD kit's cellular control is advantageous in that it allows assessment of sample quality, in contrast to the Altona kit, which includes only an inhibition control. However, when the input volume of the sample is 200 µL, the BD assay yields a high number of uninterpretable results. This ensures users do not deliver false negative results, but has the disadvantage of having to take another sample from the patient and delaying the results. Additional experiments conducted at the CHUGA Virology Laboratory on a little subset of two samples (insufficient volume prevented testing of the others) demonstrated that increasing the input volume to 400 µL improved assay performance (data not shown). As both input volumes are authorized by the manufacturer, we strongly recommend using 400 µL. It should also be noted that the Abbott kit includes both an extraction control and an inhibition control.

From a practical implementation standpoint, the Altona assay is run in batches on a 96-well plate, with the extraction and amplification steps performed on two separate instruments. The BD MAX System integrates extraction, followed by amplification without human intervention, but it still operates in batches (two runs of 24 samples, which can be processed independently). In contrast, the Alinity m System offers a random-access platform. In our experience, the main advantage of the Abbott and Altona assays is their low rate of uninterpretable results. However, the Abbott kit has a short on-board reagent stability period on the Alinity m (12 days), which may be a limiting factor for laboratories receiving few MPV samples or operating outside of outbreak periods. Conversely, the BD assay benefits from lyophilized reagents with a long shelf life. It should also be noted that the Altona workflow may increase the time to results (approximately 5 h) compared with BD and Abbott (approximately 3 h) (Table 5).

The main limitation of this study is the small number of samples and the absence of replicates for each PCR assay due to the limited sample volume available. Indeed, in the automated platforms used to validate the various assays, the volume of sample required was substantial, and these systems do not allow re-use of the DNA extract. Because of this low number of replicates, the limit of detection and the confidence interval cannot be reliably calculated; however, the lowest concentration detected can still be determined.

This study showed that there are no major differences among the three commercial systems, all of which can be used with confidence. The kits should be operated in strict

**TABLE 5** Workflow characteristics

| | BD[a] | Abbott[b] | Altona[c] |
|---|---|---|---|
| Turnaround time | ~3 hours | ~3 hours | ~7 hours |
| Batch or Random access | Batch | Random access | Batch |
| Equipment (Extraction/PCR) | Extraction and PCR | Extraction and PCR | Extraction and PCR independently |
| Reagent stability | Lyophilized, expiration date | 12 days on board | Expiration date |
| Sample collection volume | 200 µl – 400 µL | 600 µl | 200 µl |

[a]BD, VIASURE monkeypox virus Real Time PCR Reagents on the BD MAX System.
[b]Abbott, MPXV AMP Kit on the Alinity m system.
[c]Altona, RealStar Zoonotic *Orthopoxvirus* PCR Kit 1.0 on the Argene solution.

accordance with the manufacturer's instructions and with appropriate quality controls to ensure a reliable and certified result for the patient.

## Highlights

This study evaluated three commercial systems for MPV DNA detection: the VIASURE monkeypox virus Real-Time PCR Reagents on the BD MAX System; the MPXV AMP Kit on the Alinity m instrument; and the Altona RealStar Zoonotic *Orthopoxvirus* PCR Kit 1.0 on the Argene solution platforms (EMAG, ESTREAM [Biomérieux], and LC480 [Roche]).

The reference method was the one used by the French National Reference Center for orthopoxviruses.

The sensitivity and specificity of all three commercial systems were 100% on the samples tested.

The three commercial systems yielded lower Ct values compared to the method used by the French National Reference Center.

## ACKNOWLEDGMENTS

Laboratory tests were kindly provided by Abbott Laboratories and Becton Dickinson. Companies had no role in data collection and interpretion.

## AUTHOR AFFILIATIONS

[1]Laboratoire de Virologie, IBP, CHU Grenoble Alpes, La Tronche, France
[2]Aix-Marseille Univ, MEPHI, Marseille, France
[3]Univ. Grenoble Alpes, CNRS, CEA, IRIG IBS, Grenoble, France
[4]CNR *Orthopoxvirus*, Unité de Virologie, IRBA, Brétigny-sur-Orge, France

## PRESENT ADDRESS

Julien Andréani, Assistance Publique des Hôpitaux de Marseille (AP-HM), Institut Hospitalo-Universitaire Méditerranée Infection, Marseille, France

## AUTHOR ORCIDs

Julien Andréani  http://orcid.org/0000-0002-8630-4763
Aurélie Truffot  http://orcid.org/0000-0002-7991-1733
Sylvie Larrat  http://orcid.org/0000-0001-7817-6602
Julien Lupo  http://orcid.org/0000-0002-6755-3115
Patrice Morand  http://orcid.org/0000-0002-7456-0858
Olivier Ferraris  http://orcid.org/0000-0002-1193-5835
Raphaële Germi  http://orcid.org/0000-0001-7600-5930

## AUTHOR CONTRIBUTIONS

Marie Usal, Data curation, Formal analysis, Investigation, Methodology, Project administration, Validation, Writing – original draft | Julien Andréani, Data curation, Formal analysis, Investigation, Methodology, Project administration, Resources, Software, Validation, Writing – original draft | Aurélie Truffot, Formal analysis, Investigation, Methodology, Resources, Software, Supervision, Validation, Writing – original draft, Writing – review and editing | Capucine Chevalier, Data curation, Formal analysis, Investigation, Methodology, Resources, Supervision, Validation, Writing – review and editing | Sylvie Larrat, Conceptualization, Data curation, Formal analysis, Funding acquisition, Investigation, Methodology, Project administration, Resources, Writing – review and editing | Julien Lupo, Conceptualization, Funding acquisition, Methodology, Project administration, Resources, Software, Validation, Writing – review and editing | Patrice Morand, Investigation, Methodology, Project administration, Resources, Software, Validation, Writing – review and editing | Olivier Ferraris, Conceptualization, Data

curation, Formal analysis, Investigation, Methodology, Project administration, Resources, Validation, Writing – review and editing | Raphaële Germi, Conceptualization, Data curation, Formal analysis, Funding acquisition, Investigation, Methodology, Project administration, Supervision, Validation, Writing – original draft, Writing – review and editing

## ETHICS APPROVAL

This study involves data and samples from human participants in CHUGA Grenoble University Hospital and in IRBA, according to current French regulations.

The CHUGA investigator (R.G.) has signed a commitment to comply with Reference Methodology no. 004 issued by French Authorities (CNIL). The IRBA is the national reference laboratory for orthopoxviruses, designated by the French Ministry of Health (through the "Arrêté du 7 mars 2017 fixant la liste des centres nationaux de référence pour la lutte contre les maladies transmissibles") to process human samples they collect for identification and characterization of MPV.

## ADDITIONAL FILES

The following material is available online.

Open Peer Review

**PEER REVIEW HISTORY (review-history.pdf).** An accounting of the reviewer comments and feedback.

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
