## [Reviewer comments · Microbiology Spectrum]

Microbiology Spectrum

Evaluation and comparison of three qPCR commercial assays and three automated platform for the detection of monkeypox virus DNA.

Marie Usal, Julien Andreani, Aurélie Truffot, Capucine Chevalier, Sylvie LARRAT, Julien LUPO, Patrice MORAND, Olivier Ferraris, and raphael GERMI

Corresponding Author(s): raphael GERMI, CHU Grenoble

Review Timeline:

Submission Date:	November 20, 2025
Editorial Decision:	January 13, 2026
Revision Received:	March 9, 2026
Accepted:	March 19, 2026

Editor: Marisa Nielsen

Reviewer(s): Disclosure of reviewer identity is with reference to reviewer comments included in decision letter(s). The following individuals involved in review of your submission have agreed to reveal their identity: Gundallahalli Bayyappa Manjunathareddy (Reviewer #1)

Transaction Report:

DOI: <https://doi.org/10.1128/spectrum.03680-25>

Re: Spectrum03680-25 (**Evaluation and comparison of three qPCR commercial assays and three automated platform for the detection of monkeypox virus DNA.**)

Dear Prof. raphael GERMI:

Thank you for the privilege of reviewing your work. Below you will find my comments, instructions from the Spectrum editorial office, and the reviewer comments.

Revision Guidelines

Sincerely,
Marisa Nielsen
Editor
Microbiology Spectrum

Reviewer #1 (Comments for the Author):

Major Comments

1. Sample Size and Analytical Robustness

The total number of samples analyzed ($n = 52$) is relatively limited, and technical replicates were not performed due to sample volume constraints. While this limitation is acknowledged in the Discussion, it would be helpful to emphasize earlier that the

study primarily assesses comparative analytical performance rather than establishing definitive sensitivity metrics such as limit of detection (LOD) with confidence intervals. This clarification would appropriately frame the strength of the conclusions.

2. Interpretation of Ct Value Differences

The manuscript reports that the commercial assays generally produced lower Ct values than the NRC reference assay. While this observation is interesting, Ct values are assay-dependent and influenced by multiple factors, including extraction efficiency, target gene selection, chemistry, and calibration standards. Lower Ct values should therefore not be interpreted as inherently superior analytical performance. A brief qualifying statement in the Results or Discussion would help prevent over-interpretation.

3. Reference Method Considerations

The NRC in-house assay is appropriately used as a reference standard given its national reference status. However, additional methodological detail or a brief discussion of its validation history would strengthen confidence in its use as a comparator and clarify potential sources of systematic bias.

4. Specificity and Orthopoxvirus Detection

The Altona RealStar® assay is designed to detect non-variola Orthopoxviruses and therefore correctly identifies cowpox and vaccinia viruses. This expected behavior is explained in the manuscript but introducing this information earlier-particularly in the Methods or Results-would improve clarity for readers less familiar with Orthopoxvirus diagnostics.

5. Internal and Cellular Control Performance

The higher number of uninterpretable results observed with the BD MAX{trade mark, serif} assay due to cellular control failure, particularly in MPXV-negative samples, represents an important operational finding. While the issue is discussed, a clearer emphasis on its potential impact in routine diagnostic settings-especially in low-prevalence contexts-would enhance the practical value of the study. The observation that increasing input volume improved performance is useful, but readers would benefit from a brief acknowledgment that these data were not systematically evaluated.

Minor Comments

1. Minor grammatical and typographical errors are present and should be corrected during final editing.
2. Abbreviations for the National Reference Center (NRC/NCR) should be standardized throughout the manuscript.
3. Figures and tables should be consistently referenced in sequential order in the text.
4. The discussion of workflow characteristics (turnaround time, batch versus random access, reagent stability) is informative and could be further strengthened by inclusion of a concise summary table.
5. The ethics and regulatory compliance section is thorough but could be slightly condensed for readability.

Reviewer #2 (Comments for the Author):

This study evaluated three platforms for the molecular diagnosis of mpox in comparison with the reference diagnostic method used by the French National Laboratory: BD MAX{trade mark, serif} System (VIASURE Monkeypox virus Real-Time PCR Reagents), Abbott Alinity m® System (MPXV AMP Kit), and Altona RealStar® Zoonotic Orthopoxvirus PCR Kit 1.0 (Argene/EMAG® + ESTREAM® + LightCycler® 480 II)

Although the manuscript's aims are relevant, several points require clarification and should be addressed in the revised version before the work can be considered further for publication. Below, I outline questions and suggestions intended to strengthen the manuscript and improve its clarity.

Major points:

- 1- The authors characterized 32 clinical specimens with consistent positivity across the four methods. However, the information regarding the clinical samples is insufficiently described in the manuscript. I kindly request the inclusion of basic metadata for the clinical specimens in a dedicated table, providing details such as sample type (e.g., lesion swab, crust/scab, nasal swab-as mentioned by the authors), sex, age, result (positive), date of collection, and the corresponding classification (Clade I or Clade II).
- 2- In line with the previous suggestion, the clinical samples analyzed had been previously tested using the National Laboratory's molecular method. Because these specimens originated from a routine diagnostic setting, it is plausible that the National Laboratory testing was performed earlier and across different dates. This timeline is critical and, should be explicitly stated, particularly in a study comparing diagnostic methods. Moreover, samples may have undergone one or more freeze cycles between the Central Laboratory test and the subsequent assays, which can affect nucleic acid integrity and, consequently, influence results such as Ct values. It is also essential to clarify whether the compared assays were performed using the same extraction eluate or via independent extractions, and whether any extraction protocols differed across platforms. These details should be described more clearly and comprehensively than in the current manuscript to ensure that the comparisons are appropriately supported. I recommend expanding the proposed metadata table to include (at minimum) the NRC diagnosis date, the number of freeze cycles, and extraction-related information (including whether a shared eluate was used or separate extractions were performed, and any protocol differences).
- 3- More details should be provided on the viral culture supernatants tested. Reporting only that vaccinia or cowpox were positive is superficial. Cowpox viruses display the highest genetic diversity among Orthopoxviruses and cluster into several distinct clades. Likewise, vaccinia viruses comprise multiple vaccine lineages and vary substantially in genome size. Therefore, for all laboratory viral supernatants used in this study, please provide: (i) the strain/lineage identity; (ii) where applicable, whether the material was obtained from ATCC or another biobank, including the corresponding catalog/accession number; and (iii) if the material represents an isolate previously characterized in the literature, the appropriate reference.
- 4- I am not fully convinced that a direct comparison of Ct, Δ Ct, and especially the use of Bland-Altman applied to comparisons

across different methodologies is appropriate in this manuscript, as it is likely to introduce substantial bias in the interpretation. Comparing Ct values across assays targeting different molecular regions is methodologically fragile and can be misleading, because Ct depends on multiple assay and platform-specific factors, including the region amplified, primer and probe efficiency, assay chemistry and instrument detection settings (baseline/threshold, software algorithms and configuration), among others. Therefore, I would recommend that the manuscript treat Ct as not directly comparable across assays and focus the analysis on more robust metrics such as positive/negative, limit of detection. Ct values can still be reported for transparency and used for within-assay comparisons, where analytical conditions are equivalent.

5- I was unclear about the NRC "in-house" protocol. Does this refer to the assay based on the primer/probe set described by Li et al., 2010 (reference 22)? If so, I suggest avoiding the label "in-house," as this protocol is widely adopted and is arguably one of the most used qPCR assays worldwide for MPV detection. Please clarify this point and, if applicable, describe it explicitly as the Li et al. method.

Minor points:

- Please revise the manuscript to ensure strict ICTV-compliant taxonomy (<https://ictv.global/>) and virus naming throughout-use italics only for taxonomic ranks (family/genus/species), and use non-italic lowercase for virus names as common entities. Also update and standardize the current species names/abbreviations for Orthopoxvirus monkeypox (MPV), Orthopoxvirus vaccinia, and , Orthopoxvirus cowpox consistently across the text, tables, and figures.
- NRC vs. NCR: the abbreviation appears swapped/inverted in several places throughout the manuscript.
- Could the authors clarify the six uninterpretable BD results by reporting the total number tested (and the invalid rate), specifying which internal control failed and the BD interpretation criteria? Additionally, were there any differences in the BD extraction workflow or input volume for these samples compared with the other methods?
- The authors conclude 100% sensitivity/specificity; however, the total panel includes only 52 samples (32 positive and 20 negative). With this sample size, the confidence intervals are necessarily wide; therefore, statements such as "excellent analytical performance" should be more cautious, and the corresponding 95% confidence intervals (or at least an explicit acknowledgement of this imprecision) should be provided. This concern is further compounded by the lack of replicate testing, which limits assessment of assay precision/reproducibility and reduces the robustness of performance estimates.

Dear Editor, dear Reviewers,

First of all, we would like to thank the reviewers for their comments, which helped us improve our manuscript.

Please find attached our point-by-point response to the comments.

We hope that these responses address your questions and requests.

If this is not the case, we remain at your disposal to provide any additional information.

Reviewer #1 (Comments for the Author):

Major Comments

1. Sample Size and Analytical Robustness

The total number of samples analyzed ($n = 52$) is relatively limited, and technical replicates were not performed due to sample volume constraints. While this limitation is acknowledged in the Discussion, it would be helpful to emphasize earlier that the study primarily assesses comparative analytical performance rather than establishing definitive sensitivity metrics such as limit of detection (LOD) with confidence intervals. This clarification would appropriately frame the strength of the conclusions.

We specified this information at the beginning of the abstract (line 30) and at the end of the introduction (line 108).

2. Interpretation of Ct Value Differences

The manuscript reports that the commercial assays generally produced lower Ct values than the NRC reference assay. While this observation is interesting, Ct values are assay-dependent and influenced by multiple factors, including extraction efficiency, target gene selection, chemistry, and calibration standards. Lower Ct values should therefore not be interpreted as inherently superior analytical performance. A brief qualifying statement in the Results or Discussion would help prevent over-interpretation.

This statement has been added in the “discussion” section (line 323-325).

3. Reference Method Considerations

The NRC in-house assay is appropriately used as a reference standard given its national reference status. However, additional methodological detail or a brief discussion of its validation history would strengthen confidence in its use as a comparator and clarify potential sources of systematic bias.

The qPCR method used at the NRC is now described in greater detail in the “methods” section (lines 155-161).

4. Specificity and Orthopoxvirus Detection

The Altona RealStar® assay is designed to detect non-variola Orthopoxviruses and therefore correctly identifies cowpox and vaccinia viruses. This expected behavior is explained in the manuscript but introducing this information earlier-particularly in the Methods or Results-would improve clarity for readers less familiar with Orthopoxvirus diagnostics.

This information has been added at the end of the introduction (line 112-113).

5. Internal and Cellular Control Performance

The higher number of uninterpretable results observed with the BD MAX{trade mark, serif} assay due to cellular control failure, particularly in MPXV-negative samples, represents an important operational finding. While the issue is discussed, a clearer emphasis on its potential impact in routine diagnostic settings-especially in low-prevalence contexts-would enhance the

practical value of the study. The observation that increasing input volume improved performance is useful, but readers would benefit from a brief acknowledgment that these data were not systematically evaluated.

Thank you for this comment which helped improve our discussion (lines 357-365).

Minor Comments

1. Minor grammatical and typographical errors are present and should be corrected during final editing.

2. Abbreviations for the National Reference Center (NRC/NCR) should be standardized throughout the manuscript.

Thank you for your vigilance. All occurrence of “NCR” have been corrected to “NRC”.

3. Figures and tables should be consistently referenced in sequential order in the text.

We are very sorry, but we could not identify the issue.

4. The discussion of workflow characteristics (turnaround time, batch versus random access, reagent stability) is informative and could be further strengthened by inclusion of a concise summary table.

Table 5 has been added and cited in line 378.

5. The ethics and regulatory compliance section is thorough but could be slightly condensed for readability.

As requested, we have condensed the paragraph.

Major points:

1- The authors characterized 32 clinical specimens with consistent positivity across the four methods. However, the information regarding the clinical samples is insufficiently described in the manuscript. I kindly request the inclusion of basic metadata for the clinical specimens in a dedicated table, providing details such as sample type (e.g., lesion swab, crust/scab, nasal swab- as mentioned by the authors), sex, age, result (positive), date of collection, and the corresponding classification (Clade I or Clade II).

Thank you for your comment. We would have liked to provide a more detailed answer. However, French regulations are very strict, and the regulatory procedures we followed do not allow us to provide more specific information about the patients. As the study submitted to the ethics committee focused on a comparison of methods, we did not request authorization to use patients' clinical data or identifying information such as the date of sampling. We are therefore only authorized to use data related to the nature of the samples and the viruses they contain (e.g., matrix, viral load). However, we have provided additional information in the text regarding samples storage instead of date of collection (line 147-150).

Moreover, unlike to 2024 outbreak, clades were not systematically determined in the routine practice during the 2022 outbreak, and we therefore do not have this information for the clinical samples used in this study.

2- In line with the previous suggestion, the clinical samples analyzed had been previously tested using the National Laboratory's molecular method. Because these specimens originated from a routine diagnostic setting, it is plausible that the National Laboratory testing was performed earlier and across different dates. This timeline is critical and, should be explicitly stated, particularly in a study comparing diagnostic methods.

Moreover, samples may have undergone one or more freeze cycles between the Central Laboratory test and the subsequent assays, which can affect nucleic acid integrity and, consequently, influence results such as Ct values.

We have provided additional information in the text regarding sample storage time and the number of freeze-thaw cycles (line 147-150).

It is also essential to clarify whether the compared assays were performed using the same extraction eluate or via independent extractions, and whether any extraction protocols differed across platforms. These details should be described more clearly and comprehensively than in the current manuscript to ensure that the comparisons are appropriately supported.

All analyses, with all methods were performed independently, i.e., directly on the primary collected samples. No DNA extract were reused. In addition, the automated platforms do not allow retrieval of DNA extracts.

I recommend expanding the proposed metadata table to include (at minimum) the NRC diagnosis date, the number of freeze cycles, and extraction-related information (including whether a shared eluate was used or separate extractions were performed, and any protocol differences).

Please, see comment 1.

3- More details should be provided on the viral culture supernatants tested. Reporting only that vaccinia or cowpox were positive is superficial. Cowpox viruses display the highest genetic diversity among Orthopoxviruses and cluster into several distinct clades. Likewise, vaccinia viruses comprise multiple vaccine lineages and vary substantially in genome size. Therefore, for all laboratory viral supernatants used in this study, please provide: (i) the strain/lineage identity; (ii) where applicable, whether the material was obtained from ATCC or another biobank, including the corresponding catalog/accession number; and (iii) if the material represents an isolate previously characterized in the literature, the appropriate reference.

We have added the requested information about the strains in lines 128-134 and the reference describing the MPV strain isolated in the NRC laboratory.

4- I am not fully convinced that a direct comparison of Ct, Δ Ct, and especially the use of Bland-Altman applied to comparisons across different methodologies is appropriate in this manuscript, as it is likely to introduce substantial bias in the interpretation. Comparing Ct values across assays targeting different molecular regions is methodologically fragile and can be misleading, because Ct depends on multiple assay and platform-specific factors, including the region amplified, primer and probe efficiency, assay chemistry and instrument detection settings (baseline/threshold, software algorithms and configuration), among others. Therefore, I would recommend that the manuscript treat Ct as not directly comparable across assays and focus the analysis on more robust metrics such as positive/negative, limit of detection. Ct values can still be reported for transparency and used for within-assay comparisons, where analytical conditions are equivalent.

Thank you for this relevant comment. We agree that the Bland-Altman analysis is not fully appropriate for our study; therefore, we removed the graph and its mention in the text (“material and methods” and “results” section)

As also recommended by reviewer 1 we added a comment in the “discussion” section (line 323-325), to avoid overinterpretation of the Ct values.

5- I was unclear about the NRC "in-house" protocol. Does this refer to the assay based on the primer/probe set described by Li et al., 2010 (reference 22)? If so, I suggest avoiding the label "in-house," as this protocol is widely adopted and is arguably one of the most used qPCR assays worldwide for MPV detection. Please clarify this point and, if applicable, describe it explicitly as the Li et al. method.

Thank you for your comment, which helped clarify the manuscript. The qPCR method used at the NRC was that described by Li et al. We have removed all occurrence of the term “in-house” (which we had incorrectly used to indicate that the test was not commercial), throughout the text, tables and figures. We have also described the method used in NRC in greater detail in the “method” section (lines 155-162).

Minor

points:

- Please revise the manuscript to ensure strict ICTV-compliant taxonomy (<https://ictv.global/>) and virus naming throughout-use italics only for taxonomic ranks (family/genus/species), and use non-italic lowercase for virus names as common entities. Also update and standardize the current species names/abbreviations for Orthopoxvirus monkeypox (MPV), Orthopoxvirus vaccinia, and , Orthopoxvirus cowpox consistently across the text, tables, and figures.

Thank you for your comment. We hope that all the virus naming are now correct.

- NRC vs. NCR: the abbreviation appears swapped/inverted in several places throughout the manuscript.

Thank you for your vigilance. All occurrence of “NCR” have been corrected to “NRC”.

- Could the authors clarify the six uninterpretable BD results by reporting the total number tested (and the invalid rate), specifying which internal control failed and the BD interpretation criteria? Additionally, were there any differences in the BD extraction workflow or input volume for these samples compared with the other methods?

Thank you for this comment which significantly helped improve the manuscript.

There was 5 uninterpretable BD results, as reported in the “results” section and shown in the table 4, and not 6 as stated in the “discussion”. This has been corrected in the text (line 350).

Upon rereading the manuscript, we understand the reviewer's confusion. We have attempted to clarify the controls results in Table 4 and in the “results” section, and we hope that they are now clearer (line 254-275).

The BD interpretation criteria are presented in the table below. Given the improvements made to the text, we believe it may not be necessary to include this table in the article, but we would be happy to add it if the reviewers consider it useful.

BDMax			
Ct MPXV	Ct Cellular Control	Interpretation	Complementary information
0	< 35	To check	no Ct value calculated by the software with the specified Threshold (200) Amplification curve must be checked manually
-1	< 35	Negative	
< 40	whatever the result	Positive	IC detection not necessary => a high copy number of the target can cause preferential amplification of pathogen target
-1	-1	UNR - unresolved result	False negative possible => sample with few cells

There is a difference in the input volume between BD (200-400µl) and Alinity (600µl), Altona (200µl) and NRC (200µl). This information has been added in the new table 5 requested by the reviewer 1;

However, for each technology, the same input volume was used for all sample tested.

- The authors conclude 100% sensitivity/specificity; however, the total panel includes only 52 samples (32 positive and 20 negative). With this sample size, the confidence intervals are necessarily wide; therefore, statements such as "excellent analytical performance" should be more cautious, and the corresponding 95% confidence intervals (or at least an explicit acknowledgement of this imprecision) should be provided. This concern is further compounded by the lack of replicate testing, which limits assessment of assay precision/reproducibility and reduces the robustness of performance estimates.

We agree that, given the small sample size, the wording used was not appropriate. We have therefore modified the text in the “abstract”, in the “importance” section, in the “highlight”, in the “results” and in the “discussion” section to replace the term “excellent” with more caution wording and to clarify that the 100% performance refers only to the samples tested

Summary: line 43 and 47

Importance: line 60

Highlight: line 73

Results: line 202

Discussion: lines 294, 300-303 and 304

We also agree that increasing the number of replicates would be beneficial. However, as different automated platforms were used to evaluate the various assays, the volume of sample required was substantial, and these systems do not allow reuse of the DNA extracts. This therefore constitutes an additional limitation of our study. These aspects have been reported and highlighted in line 381-383 of the revised manuscript.

Re: Spectrum03680-25R1 (**Evaluation and comparison of three qPCR commercial assays and three automated platform for the detection of monkeypox virus DNA.**)

Dear Prof. raphaelle GERMI:

Your manuscript has been accepted, and I am forwarding it to the ASM production staff for publication. Your paper will first be checked to make sure all elements meet the technical requirements. ASM staff will contact you if anything needs to be revised before copyediting and production can begin. Otherwise, you will be notified when your proofs are ready to be viewed.

Sincerely,
Marisa Nielsen
Editor
Microbiology Spectrum

Reviewer #1 (Comments for the Author):

The corrections and reply from authors are satisfactory